# A New Method of Extracting *Polygonatum sibiricum* Polysaccharide with Antioxidant Function: Ultrasound-Assisted Extraction-Deep Eutectic Solvents Method

**DOI:** 10.3390/foods12183438

**Published:** 2023-09-15

**Authors:** Chaoqun Sun, Guodong Wang, Jing Sun, Jiyong Yin, Jian Huang, Zizi Li, Di Mu, Menglu He, Tingting Liu, Jiali Cheng, Hanchen Du, Yan Chen, Wenjie Qu

**Affiliations:** Key Laboratory of Trace Element Nutrition of National Health Commission of the People’s Republic of China, National Institute for Nutrition and Health, Chinese Center for Disease Control and Prevention, Beijing 100050, China; sunchaoqun97@163.com (C.S.); wanggd@ninh.chinacdc.cn (G.W.); sunjing@ninh.chinacdc.cn (J.S.); huangjian@ninh.chinacdc.cn (J.H.); lizz@ninh.chinacdc.cn (Z.L.); mudi@ninh.chinacdc.cn (D.M.); heml@ninh.chinacdc.cn (M.H.); liutt@ninh.chinacdc.cn (T.L.); chengjl@ninh.chinacdc.cn (J.C.); duhanchen9805@sina.com (H.D.); chenyan7341@163.com (Y.C.); qwj370612@163.com (W.Q.)

**Keywords:** *Polygonatum sibiricum* Polysaccharide (PsP), ultrasound-assisted extraction-deep eutectic solvents (UAE-DESS), antioxidant function, response surface method (RSM)

## Abstract

*Polygonatum sibiricum* Polysaccharide (PsP) with antioxidant function is the main active component of *Polygonatum sibiricum* (P.sibiricum). The currently poor extraction yield and extraction methods of PsP cannot meet the application of that in food industrial production. In this research, an ultrasound-assisted extraction-deep eutectic solvents (UAE-DESs) method, which has never been used in the PsP industry, was first used to extract PsP. The extraction conditions were optimized by the response surface method (RSM). Both the extraction yield and antioxidant function were simultaneously considered during the optimization process. The indicators of PsP’s level and antioxidant activity in vitro were used to present the extraction yield of the UAE-DESs method, the purity, and the antioxidant effect of PsP. Under the optimal conditions, which included a liquid–solid ratio of 26:1 (mL:g), extraction temperature of 80 °C, ultrasonic time of 51 min, and ultrasonic power of 82 W, the PsP extraction yield could reach (43.61 ± 0.09)%, which was obviously higher than single DESs (33.81%) and UAE (5.83%), respectively, and the PsP appeared favorably antioxidant function. This research proposed an efficient extraction method for PsP, filled the basic research gap, and further improved the development of PsP as a dietary supplement with antioxidant function in the food industry.

## 1. Introduction

*Polygonatum sibiricum* (P.sibiricum) (see Abbreviations) (*Polygonati rhizoma*), a common traditional Chinese food, mainly grows in Southern, Southwestern, and Northern China [1]. P.sibiricum has rich nutrients, including polysaccharides and other nutritional active ingredients, which is a kind of food with important nutritional functions, so it is widely cultivated and applied [2].

Plant polysaccharide is macromolecular carbohydrates formed by chemical bonds of many monosaccharide molecules. As one of the most abundant biological macromolecules in nature, it has a wide range of health-promoting effects, including antioxidant, hypoglycemic, anti-inflammatory, and anti-tumor effects [3,4,5,6,7,8]. Due to the above positive profile, they possess substantial application potential as nutritional function food. In recent years, the application of *Polygonatum sibiricum* Polysaccharide (PsP) (see Abbreviations) has also received extensive attention, and Wan et al. [9] once reported PsP had been applied to the research of functional food development in 2023, the results of which showed that PsP had the potential to be a dietary supplement.

The extraction yield and physiological activity of PsP are significantly affected by the extraction methods. The key to the development and utilization of PsP is to improve the method of extraction and purification of PsP so as to efficiently obtain functional components and preserve their activity to the fullest extent possible [10,11]. At present, the extraction methods mainly include hot water extraction, alkaline extraction, acid extraction, enzyme extraction, microwave-assisted extraction, freeze–thaw-assisted extraction, etc. [12,13,14,15]. Jing et al. [16] once compared the extraction yields of these different extraction methods for PsP. This study perfects the basic research on the extraction process of PsP and provides valuable experience for the follow-up research. The results showed that the extraction yield of each extraction method was less than 15%, although each one had its own advantages considering the PsP extraction yields could be further improved so as to meet the requirements of industrial production and application. 

Kakar et al. [17] once pointed out the main principle of extracting polysaccharides is to enhance the diffusion of polysaccharides from the matrix to the extraction solvents. 

Deep eutectic solvents (DESs) (see Abbreviations), which could enhance the diffusion of polysaccharides, are a novel type of extraction solvents that offer several advantages over conventional solvents and ionic liquids, including low price, being easy to obtain, good biodegradability, stability, and having low toxicity or being non-toxic [18,19]. Moreover, the internal structure of DESs can form hydrogen bonds with target polysaccharides and create electrostatic interactions, significantly improving the extraction efficiency of polysaccharides [20]. More and more studies have shown that the extraction with DESs of plant active components can not only significantly improve the extraction yield, but also retain the functional activity of the active substances [21,22]. DESs are characterized by their low surface tension and viscosity, which enable them to penetrate into polysaccharides more effectively and enhance extraction efficiency. Due to the differences in the high polarity and hydrophobicity in different DESs, they can extract various types of polysaccharides [18]. In the polysaccharide extraction process, the ethanol precipitation method was used to obtain polysaccharides. During the ethanol precipitation process, the DESs were dissolved in ethanol, and then removed with ethanol by centrifuge. In the extraction process of polysaccharides, DESs only played the role of extraction solvents without any complex operation. Thus, DESs present a unique advantage, such as safety, easy manipulation, and removement, in the extraction and separation of natural plants [23]. As an economical and innovative green solvent, they have been widely applied in the separation, extraction, and synthesis of food industries, based on the above characteristics [19,24]. Tang [25] once adopted DESs to extract PsP, which extraction yield could reach 33.81%.

Then, the extraction yield of the DESs method was tested to examine whether it could be further improved with the assistance of other methods so as to further meet the requirements of food industries.

Breaking down cell walls is a key mechanism by which the ultrasonic method improves extraction yield [26]. Therefore, the ultrasound method could also realize the diffusion of polysaccharides, which owes a series of advantages, such as short extraction time and simplified operation [27,28]. Jing et al. [16] once adopted the ultrasound method to extract PsP, of which the extraction yield was 5.83%. Although the extraction yield of only ultrasound is low, it is still possible to use ultrasound as an assistance condition to improve the extraction yield of other methods, because the material matrix is more easily broken, enabling the efficient and short-term release of polysaccharides under the conditions of ultrasound assistance [29]. Given this, it is worthwhile to explore the potential of combining ultrasound-assisted extraction with other techniques to further enhance extraction efficiency. However, the ultrasound may also lead to the degradation of polysaccharide structure and consequent loss of its biological activity. Therefore, the ultrasonic parameters should be optimized when used to assist other methods [30]. Whether we could combine the above two methods to further improve the extraction yield under the precondition of preserving biological activity requires research. Current research has shown that the ultrasound-assisted extraction-deep eutectic solvents (UAE-DESs) (see Abbreviations) method had never been used in extracting PsP. Therefore, the purpose of this research is to investigate the possibility of the UAE-DESs method in increasing the extraction yield and simultaneously considering the antioxidant function of PsP, so as to meet the requirements of food industrial production and the application of PsP. 

In the research, we utilized the response surface method (RSM) (see Abbreviations) to optimize the extraction conditions of obtaining the optimal extraction yield of UAE-DESs method [31]. As the research for PsP in North China was not enough, we specifically selected P.sibiricum from Beijing as the source material in our research. We hope to promote the development of PsP as a functional food, which can be further used as dietary supplements, through the below experiments. 

## 2. Materials and Methods

### 2.1. Materials

The roots of P.sibiricum were selected from Shunyi District (Beijing, China). All other chemicals were of analytical grade. All the reagents of this research can be searched from Table 1.

### 2.2. Main Instruments and Equipment

Allegra x-22 R centrifuge (Beckman coulter, Inc., Brea, CA, USA); SpectraMax I3X Enzyme marker (Molecular Devices Instruments Ltd., San Jose, CA, USA); desk centrifuge 5418 (Eppendorf, Inc., Hamburg, Germany); vortex mixing device ORTEx Genius (IKA, Inc., Staufen, Germany); digital ultrasonic cleaner (Kunshan Ultrasonic Instruments Co., Ltd., Kunshan, China); magnetic stirrers (IKA, Inc., Staufen, Germany); U-3900 spectrophotometer (Hitachi, Ltd., Tokyo, Japan); circulating water multi-purpose vacuum pump (Gongyi Yuhua Instrument Co., Ltd., Zhengzhou, China); freeze dryer (Ningbo Scientz Biotechnology Co., Ltd., Zhejiang, China); automatic part collector for numerical control drip meter (Shanghai huxi Analysis Instrument Factory Co., Ltd., Shanghai, China); gel size exclusion chromatograph (Agilent Technologies, Inc., Santa Clara, CA, USA); rotary evaporator (Shanghai xiande experimental instrument Co., Ltd., Shanghai, China); high-performance liquid chromatography (Agilent Technologies, Inc., Santa Clara, CA, USA).

### 2.3. Preparation of DESs

Choline chloride and 1,4 butanediol were mixed in a molar ratio of 1:4, then a certain amount of water was added to make the water content reach 33% [25]. The water bath was stirred until completely dissolved at 60 °C until a clear and transparent DESs was formed.

### 2.4. Extraction and Purification

The P.sibiricum after being dried to a constant weight in an oven (45 °C) was crushed through an 80-micron mesh sieve to obtain the P.sibiricum powder. We accurately weighed a precise amount of P.sibiricum powder into a centrifuge tube. Then the deep eutectic solvent was added according to the corresponding liquid-to-material ratio and we extracted PsP by ultrasound at a certain time and temperature. PsP was precipitated with 80% ethanol concentration. Samples were incubated 12 h at 4 °C, followed by centrifugation to collect PsP. The extraction yield of PsP from P.sibiricum was determined by the phenol-sulfuric acid method. The PsP concentration C (mg/mL) was obtained by parallel determination for 3 times. Extracts were deproteinized using the Sevage method. The dialysis bag with 1000 Da specification was selected for dialysis. We put PsP in a dialysis bag and put it on a magnetic agitator and rotated it in distilled water for dialysis. We changed the distilled water every 24 h for a total of 2 days. The impurities and small molecules were further removed by centrifugation after dialysis. After freeze-drying, it becomes PsP. The extraction yield can be calculated using Formula (1) [32]:R (%) = C × V/M × 100(1)

R: extraction yield, C: PsP concentration, V: dilution multiple, M: P.sibiricum quality.

### 2.5. Chemical Composition Analysis

The phenol-sulfuric acid method and Bradford’s method were used to determine the total sugar and protein content with D-glucose and bovine serum albumin serving as standards, respectively. UV spectroscopy was used to detect the protein and nucleic acid content of PsP [33].

### 2.6. Single-Factor Experiments

The effects of the liquid–solid ratio [10:1, 20:1, 30:1, 40:1, 50:1 (mL:g)], ultrasonic power (40, 60, 80, 100, 120 W), ultrasonic time (10, 20, 30, 40, 50, 70 min), and extraction temperature (40, 50, 60, 70, 80 °C) on the extraction yield of PsP and the scavenging rate of DPPH radicals were investigated, respectively. In each single-factor experiment, one contributing element was researched while keeping the other parameters constant. The unchanged values were identified as having a liquid–material ratio of 30:1 (mL:g), ultrasonic power of 80 W, ultrasonic time of 35 min, and extraction temperature of 60 °C. Each experiment was put into practice two times.

### 2.7. Response Surface Method (RSM)

According to the Box–Behnken design principle of RSM, the liquid–solid ratio A, ultrasonic power B and time C which have great influence on the extraction yield of PsP Y in single-factor experiments were selected as independent variables to optimize the extraction process.

The design factors and levels are shown in Table 2.

### 2.8. Molecular Weight Determination of PsP

We dissolved the PsP (200 mg) in 1 mL 0.2 m ammonium bicarbonate and added the sample (Bio-Gel PMel 10 column): 30 min each tube, each tube 9 mL. Then we collected each component, monitored by phenol-sulfuric acid method, made the gel chromatogram, collected each peak, and then concentrated this into a small volume (removing ammonium bicarbonate repeatedly).

The sample solution of PsP was filtered by filter membrane and determined by 20 μL high-performance gel permeation chromatography (HPGPC) TSK-gel G3000 PWXL column (column temperature 40 °C). The detector was a differential refractive detector, the flow rate was 0.5 mL/min, and the mobile phase was 0.05 m sodium sulfate. The chromatographic column used a series of different amounts of dextran (purchased by China Institute for Food and Drug Control) to draw the standard curve. 

The linear correlation was established by using log (Mw) as abscissa and retention time as ordinate [34].

### 2.9. Monosaccharide Component Detection

The monosaccharide mixed reference solution included: D-mannose, D-ribose, L-rhamnose, D-glucuronic acid, D-galacturonic acid, D-glucose, D-galactose, D-xylose, D-arabinose, and L-fucose, and we weighed each one of them precisely, respectively, and added water to prepare the monosaccharide mixed reference solution (the terminal concentration of each monosaccharide is 0.1 g/mL). We weighed a proper amount of the PsP in the ampoule bottle, added 1 mL Ultrapure (UP) (see Abbreviations) water to fully dissolve it, and then added 1 mL 4 M Trifluoroacetic acid (TFA) (see Abbreviations). The mixed solution was hydrolyzed with trifluoroacetic acid (2 M) at 110 °C for 4 h in the ampoule bottle. We opened the ampoule bottle and sucked out the sample solution, neutralized it with NaOH and volumed to 10 mL with distilled water. We took 200 μL constant volume solution and added 200 μL internal standard (2-deoxyribose), and the mixture was shaken and mixed well. Then 100 μL of the mixture was absorbed into a 1.5 mL centrifuge tube, and then we added 120 μL of 0.3 M NaOH and 0.5 M 1-phenyl-3-methyl-5-pyrazolone (PMP) (see Abbreviations), respectively. We mixed the mixture thoroughly and took a water bath at 70 °C for 1 h, avoiding light. After the water bath, the centrifuge tube was cooled down to room temperature, and then we added 100 μL 0.3 M HCI to neutralize the mixed solution to a pH of 7. We added 500 μL chloroform to the centrifuge tube for extraction, mixed thoroughly and centrifuged (7000 rpm × 5 min). The upper solution (water layer) was carefully sucked out, and we transferred it into a new centrifuge tube. This operation was repeated four times. The supernatant extracted from the last extraction was filtered into a liquid phase bottle by using a 0.22 um filter membrane, and we conducted HPLC analysis on the liquid in the liquid phase bottle. The detection conditions were as follows: the chromatographic column was Agilent Eclipse XDB-C18 (5 μm, 4.6 × 250 mm), the flow rate was 1 mL/min, the column temperature was 30 °C, the detection wavelength was 254 nm, the mobile phase was 18% the acetonitrile triethylamine solution, and the mobile phase was 60% acetonitrile triethylamine solution. We made a comparison of the HPLC maps between PsP and mixed monosaccharides reference solution, and used 2-deoxyribose as the internal standard. Finally, the molar ratio of monosaccharide composition was calculated by the internal standard method [35].

### 2.10. Antioxidant Activity In Vitro

#### 2.10.1. Scavenging Rate of DPPH Radical

Next, the 80 μmol/L DPPH methanol solution and different concentrations of PsP solution were prepared (1, 2, 4, 8, 16 mg/mL). A 0.5 mL sample was mixed with the 3.0 mL DPPH solution, shaken well, and placed at room temperature for 30 min [36]. At the same time, the vitamin C solution (10, 20, 40, 60, 80 μg/mL) was used as a positive control. The absorbance was measured at 517 nm of the spectrophotometer, and the DPPH radical scavenging rate was calculated using the following Formula (2):DPPH radical scavenging rate (%) = (1 − (A_j_ − A_i_)/A_o_) × 100(2)

A_j_: Absorbance of PsP, A_i_: Absorbance of control group, A_o_: Absorbance of blank group.

#### 2.10.2. Scavenging Rate of ABTS Radical

Then, the 7 mmol/L ABTS methanol solution and 2.45 mmol/L potassium persulfate solution were mixed at 1:1 and reacted at 4 °C for 24 h to prepare the ABTS free radical. The solution was diluted with methanol to an absorbance of 0.7 ± 0.05 (734 nm). At the same time, different concentrations of the PsP solution were prepared (1, 2, 4, 8, 16 mg/mL), 0.3 mL samples were mixed with 2.7 mL ABTS solution, shaken well, placed at room temperature for 10 min, and the absorbance was measured at 734 nm. At the same time, the vitamin C solution (10, 20, 40, 60, 80 μg/mL) was used as positive control [37]. The ABTS radical scavenging rate was calculated using the following Formula (3):ABTS radical scavenging rate (%) = (1 − (A_j_ − A_i_)/A_o_) × 100(3)

A_j_: Absorbance of PsP, A_i_: Absorbance of control group, A_o_: Absorbance of blank group.

#### 2.10.3. Scavenging Rate of Hydroxyl Radical

A 9 mmol/L FeSO_4_ solution 1.0 mL, 9 mmoL/L salicylic acid solution 1.0 mL, PsP solution 1.0 mL (1, 2, 4, 8, 16 mg/mL), and 8.8 mmol/L H_2_O_2_ solution 1.0 mL were added to the 10 mL colorimetric tube and shaken well, and then water bathed at 37 °C for 30 min. The absorbance was measured at 510 nm. At the same time, the vitamin C solution (10, 20, 40, 60, 80 μg/mL) was used as positive control [30]. The hydroxyl radical scavenging rate was calculated using the following Formula (4):Hydroxyl radical scavenging rate (%) = (1 − (A_j_ − A_i_)/A_o_) × 100 (4)

A_j_: Absorbance of PsP, A_i_: Absorbance of control group, A_o_: Absorbance of blank group.

#### 2.10.4. Ferric Ion Reducing Ability

A tripyridyltriazine solution was prepared with diluted hydrochloric acid at a 40 mmol/L concentration and its terminal concentration was 10 mmol/L. A sodium acetate solution of 300 mmol/L, FeCl_3_ solution of 20 mmol/L, and tripyridyltriazine solution were mixed according to a volume ratio of 10:1:1 to prepare the tripyridyltriazine working solution (now used). The PsP solution and ferrous sulfate standard solution of 0.1–0.6 mmol/L were prepared. Then, 0.3 mL PsP solutions (1, 2, 4, 8, 16 mg/mL) and different concentrations of the standard solution were mixed with 2.7 mL tripyridyltriazine working solution, shaken well, and water bathed at 37 °C for 10 min, and the absorbance was measured at 593 nm by spectrophotometer. At the same time, different concentrations of the vitamin C solution (0.625, 1.25, 2.5, 5, 10 μg/mL) were used as the positive control of PsP to conduct the above experiments. Taking the concentration of the ferrous sulfate standard solution as ordinate and the absorbance value as abscissa, the standard curve of ferrous sulfate was drawn [38]. Through measuring the average absorbance, the corresponding Fe_2_SO_4_ concentration was obtained on the standard curve, and the results were expressed by ferrous sulfate equivalent, namely the FRAP value (μmol/g) of the sample.

#### 2.10.5. Total Oxyradical Scavenging Capacity

We weighed the PsP solution of different concentrations (1, 2, 4, 8, 16 mg/mL) successively at 0.4 mL and put them in a 10 mL volumetric flask. Then we added 4.0 mL phosphorus molybdenum test solution (0.6 mol/L H_2_SO_4_, 28 mmol/L Na_3_PO_4_ and 4 mmol/L H_8_MoN_2_O_4_ mixed at 1:1:1 in volume ratio) to the inner side, shook the solution well, and water bathed it at 95 °C for 1.5 h, after the solution was cooled to room temperature. The absorbance was determined by ultraviolet analysis at 695 nm. At the same time, the distilled water was used as a reference solution and vitamin C solution (31.25, 62.5, 125, 250, 500 μg/mL) was used as a positive control [39]. The total antioxidant capacity was represented by the absorbance value, where a higher absorbance value indicated a stronger antioxidant capacity of the sample.

#### 2.10.6. Catalase Activity Determination In Vitro

The catalase solution with an activity unit of 200 U/mL was allocated according to a certain proportion and different concentrations of PsP solution were prepared (0.5, 1, 2, 4, 8, 16 mg/mL) at the same time. Before determination, the working solution of CAT detection was bathed in 10 min at 37 °C. The working solution of 1 mL CAT detection was taken into a 1 mL quartz colorimetric plate, and then a 35 μL sample was added and mixed for 5 s. The initial absorbance value A1 of 240 nm and the absorbance value A2 of 1 min later were measured immediately at room temperature [40]. The Δ A = A1 − A2 was calculated. CAT (U/mL) = 678 × ΔA.

### 2.11. Statistical Analysis

SPSS19.0 (IBM, Armonk, NY, USA) was used for statistical analysis. In the research, a one-way analysis of variance followed by a least significant difference (LSD) test was used in the analysis of single-factor experiments and the effects of different factors on the extraction yield for multiple comparisons. Each experiment was repeated at least three times and the data were presented as the means ± standard deviation (SD). A two-sided *p*-value < 0.05 was considered statistically significant. Statistical graphs were produced with Origin 2018 (Origin Lab Inc., Hampton, MA, USA).

## 3. Results

### 3.1. Single-Factor Experiments Results

Figure 1 presents the preliminary optimized results of the extraction conditions via single-factor experiments. As displayed in Figure 1A, the extraction yield of PsP increased gradually when the ratio of liquid-to-solid was 10:1 to 30:1 (mL:g), but decreased when the liquid-to-material ratio was 40:1 and 50:1 (mL:g). It may be due to the fact that the larger the amount of solution, the greater the concentration difference, which is beneficial to the dissolution of PsP. However, when the amounts of solvents continued to increase, the intercellular space capacity tended to be saturated and the extraction yield was no longer significantly increased. In the DPPH radical scavenging experiment, when the liquid–solid ratios were 30: 1, 40:1, and 50:1 (mL:g), the scavenging rate of DPPH radical was the highest. Therefore, the liquid–solid ratio was 20:1 (mL:g)–40:1 (mL:g) in the follow-up test.

Then the effects of ultrasonic power on extraction yield were investigated, ranging from 40 to 120 W and the results are presented in Figure 1B. As the ultrasonic power increases, the extraction yield initially increases due to the enhanced cavitation effect of the solvents facilitating the dissolution of PsP. However, the extraction yield begins to decrease gradually beyond a certain ultrasonic power level, as the increased power may cause structural decomposition of the PsP, leading to reduced extraction efficiency. In the DPPH radical scavenging experiment, the scavenging rate was the highest when the ultrasonic power was 60 W, 80 W, and 40 W. Therefore, the ultrasonic power of 40–120 W was selected for the follow-up test.

Figure 1C showed the influence of the extraction time (10–70 min) on the extraction yield. The extraction yield of PsP increased when the extraction time was 10 to 50 min and decreased when the ultrasonic time was more than 50 min. It may be because the longer the ultrasonic time, the more obvious the cavitation effect, the stronger the destructive effect on the cell wall, and the higher the extraction yield. However, when the ultrasonic time is too long, the temperature is higher, so that some PsP are decomposed, and the extraction yield is reduced. In the DPPH radical scavenging experiment, the scavenging rate is the highest when the ultrasonic time is 30, 50, and 10 min. Therefore, the ultrasonic time was chosen as 40–60 min in the follow-up test.

Figure 1D presented the results of investigating the impact of the extraction temperature (40–80 °C) on the extraction yield. The findings indicated that the extraction temperature had a significant effect on the yield of PsP, with increasing temperatures leading to higher yields. This could be attributed to the accelerated movement of solvents molecules with higher temperatures, facilitating better contact between the dry powder and solvents and thereby promoting the dissolution of PsP. In the DPPH radical scavenging experiment, the scavenging rate was the highest when the temperature was 40 °C, 60 °C, and 80 °C. However, due to the limitation of equipment, the extraction temperature is less than 90 °C, so the extraction temperature is not suitable in RSM.

### 3.2. RSM Results

#### 3.2.1. Model Establishment and Data Fitting

The results of RSM are shown in Table 3. Through the regression analysis, the regression equation of the extraction yield of PsP was Y = 43.17 − 2.12A + 0.491B + 0.5461C + 0.5877AB + 0.6192AC + 0.0909BC − 2.52A^2^ − 2.71B^2^ − 1.33C^2^ + 0.3.

#### 3.2.2. RSM Analysis

The results of the analysis of ANOVA are shown in Table 4. It could be seen from Table 4 that the model *p* < 0.0001 is extremely significant, indicating that the model is effective and reliable. The misfit *p*-value of this model is 0.3900, indicating that there is no obvious misfit factor, which could be used to analyze the test results instead of the real point of the test. The determination coefficient (R^2^) and adjustment coefficient (R_Adj_^2^) are 0.9855 and 0.9668 respectively, indicating that the model has good stability and accuracy. The results showed that A, B, C, AB, AC, BC, A2, B2, and A2 had a significant effect on the extraction yield of polysaccharides. According to the F value, the primary and secondary order of the effects of the four factors on the extraction yield of PsP is A > C > B.

#### 3.2.3. RSM Analysis and Optimization of Extraction Process

Through the contour map and 3D map, we can directly find the effect of the interaction of various factors on the extraction yield of PsP. 

Figure 2 presented the interaction effects of the liquid-to-material ratio (A, 20:1–40:1 (mL:g)), ultrasonic power (B, 40–120 W), and ultrasonic time (C, 40–60 min W) on PsP yields. As shown, all response surface curves exhibited a maximum point in these selected experimental ranges, suggesting a reasonable range of factors. As reported, the elliptical/circular contour plots reflected significant/indistinctive interaction effects between these variables. Therefore, the liquid-to-material ratio and ultrasonic power (AB), liquid-to-material ratio, and ultrasonic time (AC) exhibited obvious interaction effects (*p* < 0.05) on the PsP extraction yields, which is consistent with the conclusion in the analysis of the variance table.

#### 3.2.4. Verification Test of the Best Process Conditions

According to the optimized extraction process, four verification tests were carried out under the following conditions: liquid–solid ratio 26:1 (mL:g), extraction temperature 80 °C, ultrasonic power 82 W, and ultrasonic time 51 min. The average extraction yield was 43.61% and the error with the predicted value was 0.09%, indicating that the prediction of the model was good and the optimized process parameters were reliable.

### 3.3. Purity, Protein and Nucleic Acid Content of PsP

In this study, the PsP was purified through dialysis and deproteinization to obtain a purification yield of 98.76%. Chemical constitution analysis showed that the nucleic acid and protein contents of the PsP were 0.78 ± 0.04% and 1.11 ± 0.32%, respectively. 

The UV absorption curve of PsP was smooth and there was no absorption peak at 260 nm or 280 nm, indicating that there is no nucleic acid or protein, and the purity of PsP is high (Figure 3).

### 3.4. Composition Analysis, Molecular Weight Determination and Monosaccharide Composition of PsP

PsP-1 and PsP-2 were collected by the elution curve of P10 gel column. We collected the first peak 14–17 tubes and the second peak 23–47 tubes (Figure 4). 

According to the fact that the peak area is proportional to the content of PsP, PsP-2 is the main component of PsP.

The average molecular weight (Mw) of the PsP was estimated and the results are shown in Figure 5A. As presented, the retention time (Rt) of the PsP in the HPGPC spectrum was 15.552 min, demonstrating that the average molecular weight of the PsP was about 4.6 kDa.

The monosaccharide compositions and the relative contents were determined by HPLC analysis. The results could be seen in Figure 5B. Comparing the HPLC maps between PsP and mixed monosaccharides reference solution, it was found that five peaks in PsP were consistent with those of the mixed monosaccharides reference solution, indicating that PsP was composed of five monosaccharides. Depending on the peak position of the mixed monosaccharides reference solution, it was known that the five monosaccharides of PsP were, respectively, mannose, glucuronic acid, glucose, galactose, and xylose. By using 2-deoxyribose as the internal standard as an internal standard method, the molar ratios of the monosaccharide composition of them were 0.86, 0.30, 1.00, 0.87, and 0.05, respectively.

### 3.5. Analysis of Antioxidant Activity of PsP

#### 3.5.1. DPPH, ABTS and Hydroxyl Radical Scavenging Rate Analysis

The results of the DPPH, ABTS and hydroxyl scavenging experiments in this study showed that the scavenging effect of PsP on radicals increased gradually with concentrations ranging from 1 to 16 mg/mL, with a more pronounced effect observed at higher concentrations. In the present research, the scavenging effects of the PsP and positive control (vitamin C) on DPPH radical, ABTS radical and hydroxyl radical were determined, and the results are presented in Figure 6 and Table 5. 

#### 3.5.2. Ferric Ion Reducing Ability and Total Oxyradical Scavenging Capacity Analysis

The ferric ion reducing ability showed that there was a dose–response relationship between PsP and FRAP when the concentrations rang was between 1 and 16 mg/mL. Total antioxidant capacity experiments demonstrated a gradual enhancement of antioxidant activity with an increase of PsP concentrations (from 1 to 16 mg/mL). The effects of the PsP and positive control (vitamin C) on ferric ion reducing ability and total oxyradical scavenging capacity were determined and the results are shown in Figure 7.

#### 3.5.3. Catalase Activity In Vitro Analysis

The result showed that the antioxidant activity of catalase could be promoted by PsP at a concentration range of 1 and 4 mg/mL. However, with the continuous increase of the concentration of PsP, the promoting effect of PsP on catalase activity gradually weakened after 4 mg/mL, and even showed an inhibitory effect. The effects of the PsP and vitamin C on catalase activity in vitro were determined, and the determination results are shown in Figure 8.

## 4. Discussion

In this study, the UAE-DESs method was first applied to the extraction of PsP, and the extraction parameters were optimized by RSM. The results of our research showed that the UAE-DESs method had improved the PsP extraction yield to 43.61% and preserved its antioxidant function to the greatest extent, the purity of PsP reached 98.76%, and the contents of nucleic acid and protein only were 0.78 ± 0.04% and 1.11 ± 0.32%, respectively; thus, it can be considered that the DESs had been removed. The above results showed that the UAE-DESs method could meet the requirements of industrial production of PsP, and the PsP that was extracted from the UAE-DESs method could be developed and applied as a dietary supplement in the food industry.

The polysaccharide content is small in organisms, and it is widely distributed in biological tissues, so it is difficult to extract completely by using conventional extraction methods. Ultrasonic waves can enhance the contact between the polysaccharide and the extraction solvents; thereby, it can facilitate the dissolution of polysaccharides and improve the overall extraction efficiency. On the other hand, the formulation of DESs can be designed according to the specific type of polysaccharide, thus achieving a more selective extraction process, while preserving the natural characteristics and chemical structure of the polysaccharide [21]. Moreover, DESs comprise different hydrogen-bonded donors (1,4 butanediol, urea, glycerol, citric acid, etc.) and receptors (choline chloride, betaine, etc.), resulting in distinct characteristics among various DESs. In this research, we selected the deep eutectic solvent composed of choline chloride and 1,4 butanediol, which is based on the results of numerous comparative studies on the extraction yield of polysaccharides among different types of DESs [25]. The comparative results showed the DESs made from the mixture of choline chloride and 1,4 butanediol demonstrated a significantly higher extraction yield of polysaccharides than other DESs [25]. The main reason is a deep eutectic solvent composed of choline chloride and 1,4 butanediol may be more suitable for the extraction of PsP according to the principle of formulation design of DESs [18,21]. Given this, we selected the mixture of choline chloride and 1,4 butanediol as DESs to combine with ultrasound in the subsequent research. The result showed that the combination of ultrasound and DESs had actually achieved the improvement of the polysaccharide extraction efficiency. Additionally, the UAE-DESs method does not cause any damage to the chemical structure of the polysaccharide, ensuring the integrity of the extracted material.

Furthermore, the UAE-DESs method can significantly shorten the PsP extraction time. The conventional extraction methods of polysaccharides need to go through a long time of soaking, stirring, filtration, concentration, and other steps, which is time-consuming and laborious. The UAE-DESs method overcame these disadvantages of conventional methods through the advantage of ultrasonic wave, which achieved the fast dissolution of polysaccharides, thus shortening the overall extraction time [30].

On the third point, the UAE-DESs method has the potential to improve the purity of extracted polysaccharides. In the previous research, conventional extraction methods involve multiple steps and are often vulnerable to environmental pollution and sample damage, leading to impurities of the extracted polysaccharide. We adopted the UAE-DESs method to improve the purity of the extracted polysaccharide through shortened the extraction time by using ultrasonic wave, which could reduce the environmental pollution or sample destruction [26], and through strengthening the interaction between the DESs and polysaccharide during the extraction process [20], which could further enhance the purity of the extracted polysaccharide.

This study adopted RSM involved to the extraction yield and antioxidant effects of polysaccharide to determine the optimal extraction conditions of the UAE-DESs method, which included a liquid–solid ratio of 26:1 (mL:g), ultrasonic power of 82 W, ultrasonic time of 51 min and extraction temperature of 80 °C. Under the above conditions, the extraction yield of PsP could reach 43.61%, which is obviously higher than 33.81% if only using DESs (the solid:liquid ratio is 1:40, extraction time is 48 min, extraction temperature 89 °C) and 5.83% of only using ultrasound-assisted extraction (the solid:liquid ratio is 1:30, extraction time is 40 min, extraction temperature 60 °C, ultrasonic power is 100 W) in extracting PsP [16,25].

These optimized experiment conditions took into account both the extraction yield and DPPH radical scavenging rate. To determine the optimal extraction conditions, single-factor experiments with emphasis given to the DPPH radical scavenging rate were conducted. After the top three levels of each extraction factor of DPPH radical scavenging rate were selected, the levels of the highest extraction yield were ultimately chosen as the basis for the RSM. The experimental results indicated that the UAE-DESs method is a sustainable and effective approach for the extraction of PsP. This extraction method offers several advantages, including simple operation, high efficiency, environmental friendliness, energy efficiency, and the ability to maintain the original properties of polysaccharides. It is noteworthy that the optimized UAE-DESs method for extracting PsP did not alter the polysaccharide’s structure and composition, which not only increased the extraction yield, but also maintained the antioxidant activity of polysaccharides, realizing the overall enhancement of the utilization value of PsP. The findings of this study provide a fundamental basis for further research on the physiological activity of PsP.

In recent years, numerous studies have demonstrated the potent antioxidant activity of PsP [41]. For instance, a study revealed that PsP could considerably enhance the activities of superoxide dismutase and catalase in rats, thereby augmenting the body’s antioxidant capacity [42]. In addition, animal experiments have also proved the antioxidant activity of PsP [43]. Another study showed that PsP could increase the activity of antioxidant enzymes in cells through converted radicals into harmless molecules, such as superoxide dismutase and catalase, thus protecting cells from oxidative damage [44], sequentially improving the body’s antioxidant capacity [40]. These studies have proved the antioxidant activity of PsP, which provides an important scientific basis for its development in the field of health products and drugs.

Although in vitro antioxidant experiments have limitations, they can still be a necessary, important, and indispensable section in the whole process of the antioxidant evaluation. Therefore, there were still many scholars adopted in vitro antioxidant experiments as a necessary part of their research, which can provide a necessary exploration for the follow-up in vivo antioxidant experiments [36,38]. In our study, we utilized six different indexes to analyze and compare the antioxidant activity of PsP, so as to more comprehensively reduce the system error caused by the limitations of in vitro antioxidant experiments as possible, and extremely prove the antioxidant activity of PsP in this section. 

The results of the DPPH, ABTS, and hydroxyl scavenging experiments in this study showed that the scavenging effect of PsP on radicals increased gradually with concentrations ranging from 1–16 mg/mL, with a more pronounced effect observed at higher concentrations. The reason might be that the polysaccharide molecules of P.sibiricum have strong hydrophilicity and can form a large number of molecular chains in water [45], and these molecular chains can provide electrons to the radicals, so as to complete the unstable electrons in the radicals and make them stable, thus achieving the effect of scavenging radicals. Similarly, the ferric ion-reducing ability and total antioxidant capacity experiments demonstrated a gradual enhancement of antioxidant activity with an increase in PsP concentration (1–16 mg/mL). In ferric ion reducing ability, we selected the FRAP value to express the result. The reason is FRAP value can better show the results of antioxidant activity, and many researchers like to use this index to present the antioxidant activity [38].

In the study, our experiment about the effect of PsP on in vitro catalase activity indicated that the activity of catalase could be promoted by PsP at a concentration range of 1–4 mg/mL. However, with the continuous increase in the concentration of PsP, the promoting effect of PsP on catalase activity gradually weakened after 4 mg/mL, and even showed an inhibitory effect. This phenomenon may be attributed to the low concentration of PsP may provide an environment that can improve the activity of catalase and make the catalase more stable and active. However, the over-high concentration of PsP would alter the pH environment of catalase, and then induce structural changes, which would lead to a decrease in catalase activity. This finding suggested that PsP may enhance the catalase activity within a specific concentration range, and provided a novel pathway for future research on the effects of PsP on catalase activity. The above results of antioxidant activity experiments were consistent with others similar research [16,18].

This study did not attempt to use in vitro antioxidant experiments to replace in vivo antioxidant experiments, and the in vitro antioxidation experiment is only the starting point of our team in proving the antioxidant activity of PsP, but not the end point of that. In the follow-up studies, we successively carry out the in vivo antioxidation experiment of PsP based on the above results of in vitro antioxidant experiments, so as to more comprehensively explore the relevant mechanism and pathway of PsP antioxidation, and provide a solid foundation for PsP as a dietary supplement in the food industry.

In the above experiments, we optimized the UAE-DESs method process, and preliminarily verified and analyzed the antioxidant function, components, molecular weight, and monosaccharide composition of PsP. Through the first application of the UAE-DESs method to the extraction of PsP, the utilization efficiency of PsP in industrial production can be effectively increased after optimizing the extraction parameters. The results of this study show that our study successfully addressed the research gap for PsP and promoted the PsP application in the food industry as a natural dietary supplement. In the follow-up study, we will use the microfluidic technique to explore PsP effects in delaying cell senescence based on the previous research basis of our team [46,47].

## 5. Conclusions

In summary, the UAE-DESs method is firstly used to extract PsP from P.sibiricum, which successfully facilitates the dissolution of more PsP and maintains the PsP structure and antioxidant function. Furthermore, our study realized the optimization for the extraction conditions of the UAE-DESs method through RSM. The UAE-DESs method, which can increase the extraction yield of PsP under the premise of considering antioxidant function to the greatest extent, can realize the overall enhancement of the utilization value of PsP. This research filled the basic research gap of PsP and provided a new pathway that can improve the utilization efficiency of PsP in food industrial production.

## Figures and Tables

**Figure 1 foods-12-03438-f001:**
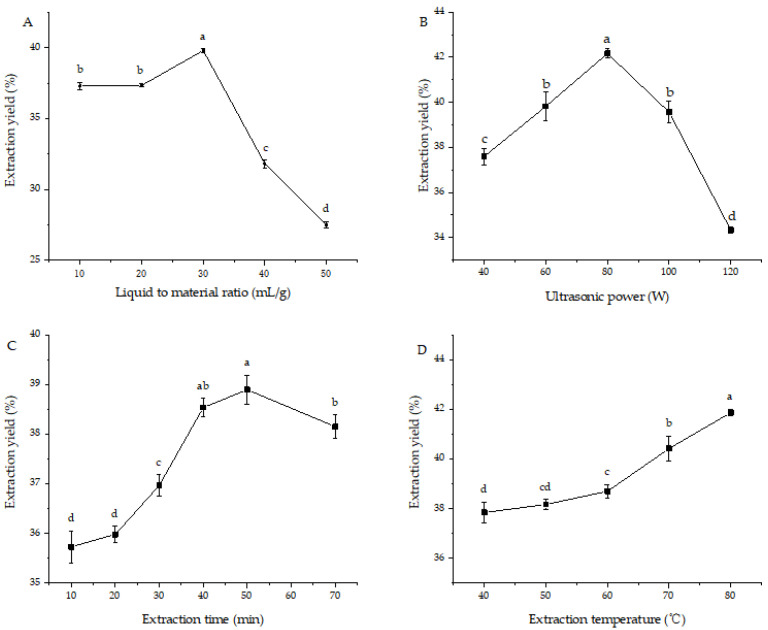
Single-factor experimental results for PsP extraction: (**A**) Liquid-to material ratio; (**B**) ultrasonic power; (**C**) extraction time; (**D**) extraction temperature. Different lowercase letters showed significant difference (*p* < 0.05).

**Figure 2 foods-12-03438-f002:**
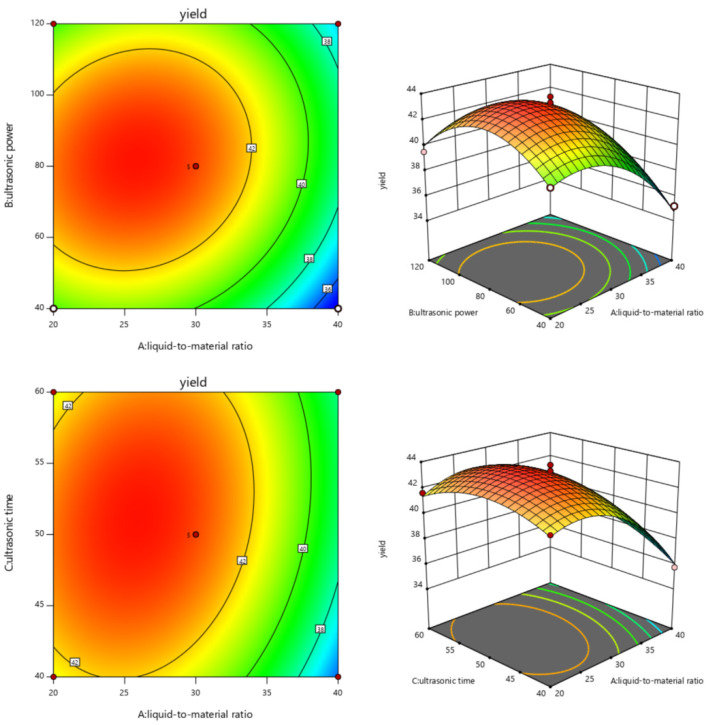
Contour plots and response surface of the PsP production with various variables, including liquid-to-material ratio, ultrasonic time, and ultrasonic power. Note: The graphic color from blue to red means that the extraction yield is from less to more, and the faster the color changes, the more significant the effect of this factor on the results.

**Figure 3 foods-12-03438-f003:**
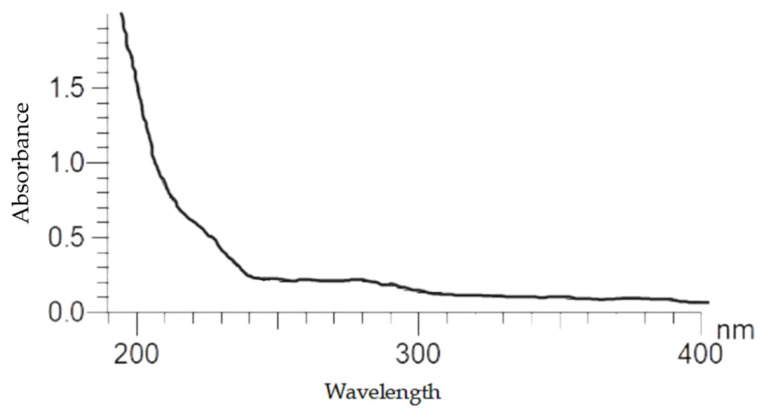
UV spectra of PsP.

**Figure 4 foods-12-03438-f004:**
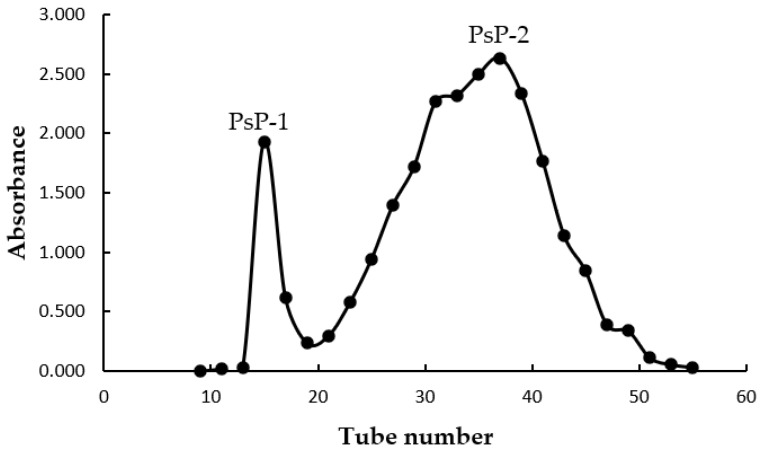
Elution curve of PsP on P10 gel column.

**Figure 5 foods-12-03438-f005:**
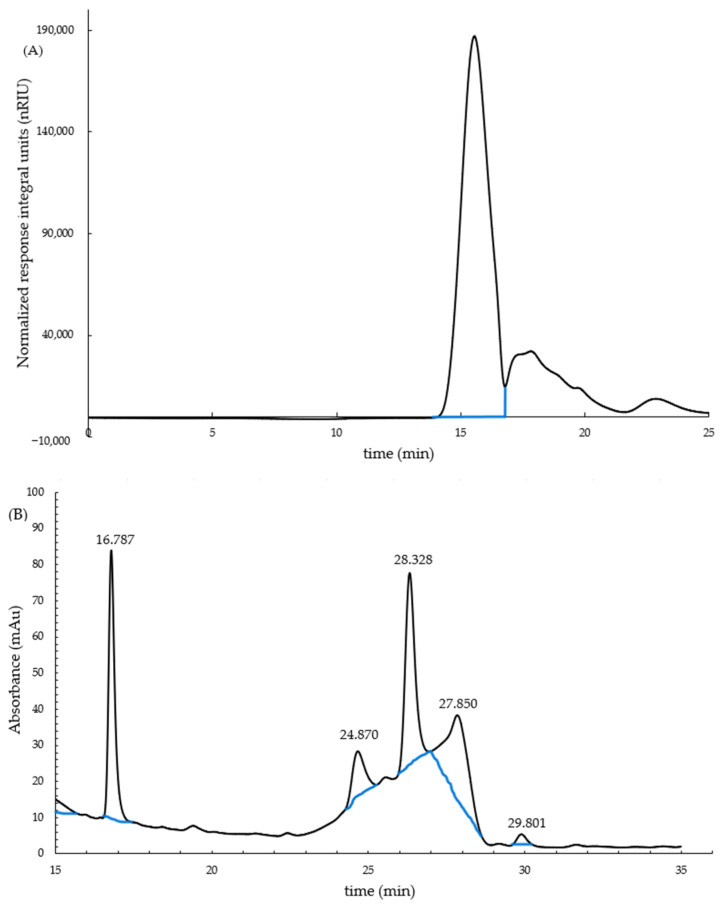
(**A**) Molecular weight distribution of PsP. (**B**) HPLC analysis the monosaccharide compositionof PsP. Note: In chronological order, the monosaccharides are mannose, glucuronic acid, glucose, galactose, and xylose.

**Figure 6 foods-12-03438-f006:**
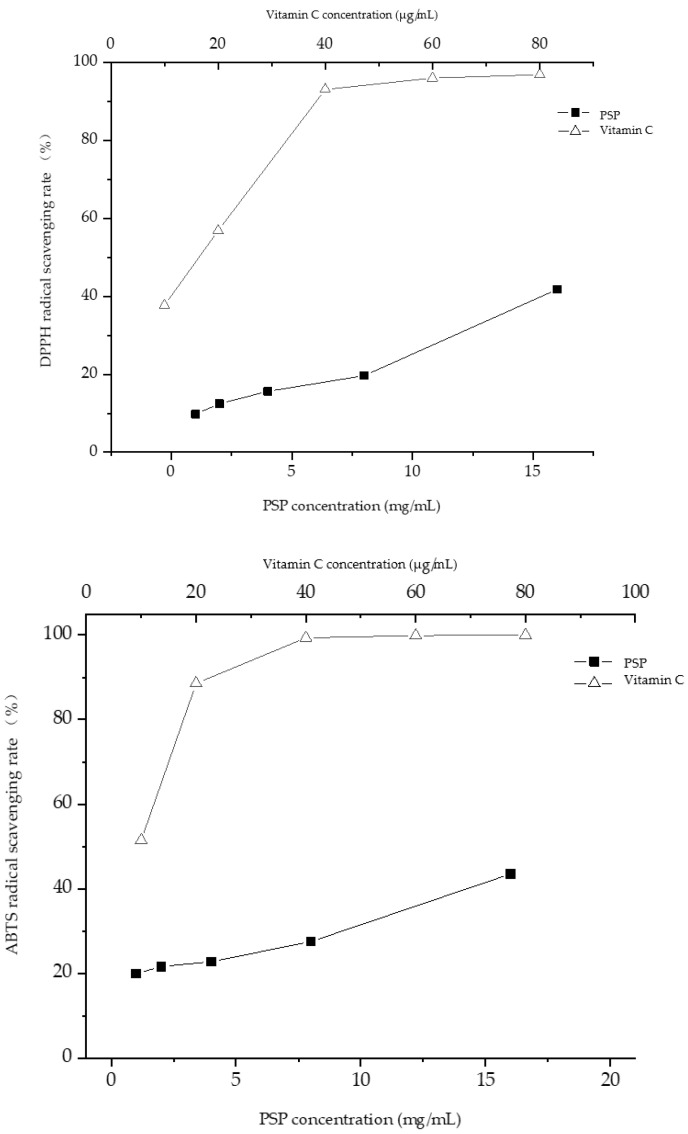
DPPH, ABTS, and hydroxyl radical scavenging rate.

**Figure 7 foods-12-03438-f007:**
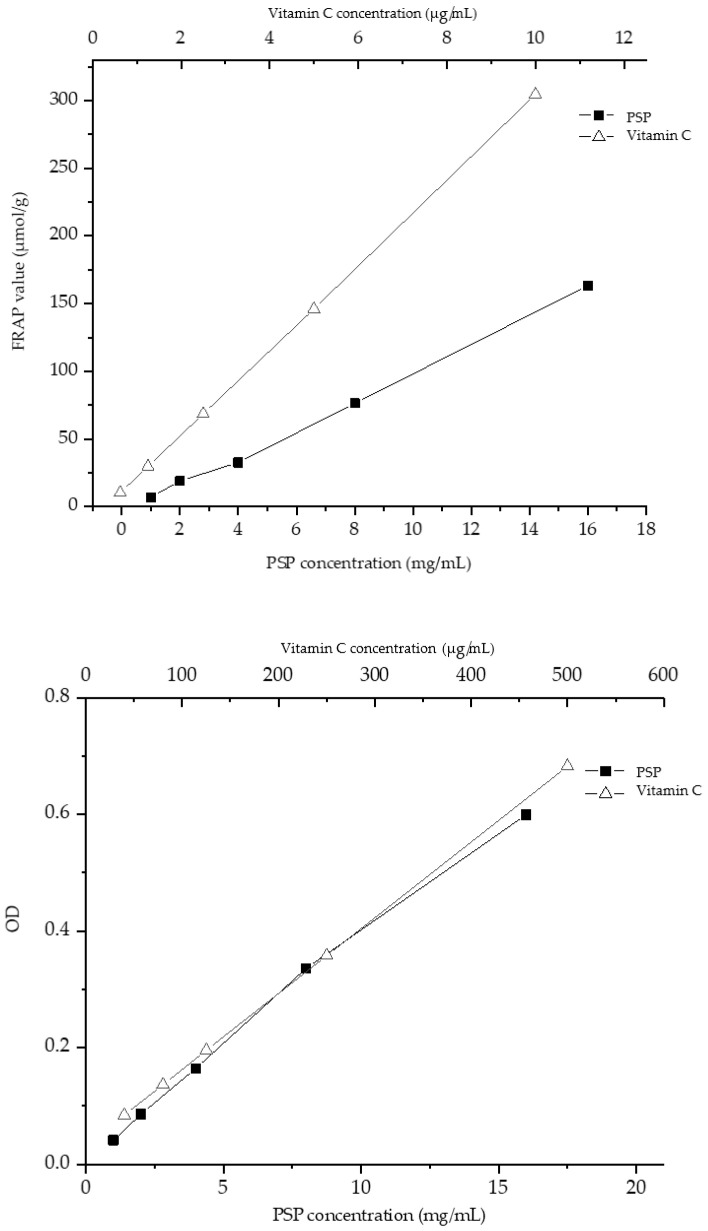
Results of ferric ion reducing ability and total oxyradical scavenging capacity.

**Figure 8 foods-12-03438-f008:**
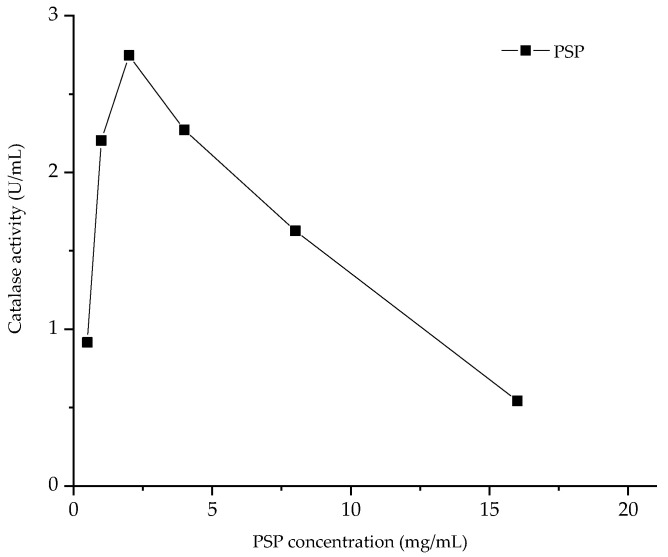
Results of catalase activity in vitro.

**Table 1 foods-12-03438-t001:** Reagents used in this research.

The Name of Product	The Code of Product	Company (City, State, Country)
2,2-diphenyl-1-picrylhydrazyl (DPPH) (see Abbreviations)	BSF220911	Shanghai Shifeng biological technology Co., Ltd. (Shanghai China)
2,2′-azino-bis(3-ethylbenzothiazoline-6-sulfonic acid) (ABTS) (see Abbreviations)	221899	Aladdin, Inc. (Shanghai, China)
Bovine serum albumin (BSA) (see Abbreviations)	WXBC7961V	Sigma Co., Ltd. (St. Louis, MO, USA)
D-Glucose anhydrous-RM	RMT13890	Beijing Manhagebio-tech, Co., Ltd. (Beijing China)
Coomassie brilliant blue G-250	727Y032	Beijing Solarbio life science, Inc. (Beijing, China)
Catalase (CAT) Assay Kit	20221008	Beijing Solarbio life science, Inc. (Beijing, China)
Catalase	1129Y021	Beijing Solarbio life science, Inc. (Beijing, China)
TSK-gel G3000 PWXL column	0008033	Guangzhou Lubex Scientific Instrument, Co., Ltd. (Guangzhou, China)
Eclipse XDB-C18	7995118-585	Agilent Technologies, Inc.(Santa Clara, CA, USA)
Bio-Gel PMel	1504140	Bio-Rad Laboratories, Inc. (Hercules, CA, USA)

**Table 2 foods-12-03438-t002:** RSM design factors and levels.

Levels	Factors		
	A Liquid-Solid Ratio(mL:g)	B Ultrasonic Power/W	C Ultrasonic Time/min
−1	20:1	40	40
0	30:1	80	50
1	40:1	120	60

**Table 3 foods-12-03438-t003:** RSM design experiment schedule and results.

Number	A/(mL:g)	B/W	C/min	Y/%
1	20:1	40	50	40.21
2	40:1	40	50	35.18
3	20:1	120	50	39.52
4	40:1	120	50	36.85
5	20:1	80	40	41.64
6	40:1	80	40	35.76
7	20:1	80	60	41.64
8	40:1	80	60	38.24
9	30:1	40	40	38.02
10	30:1	120	40	39.31
11	30:1	40	60	38.78
12	30:1	120	60	40.43
13	30:1	80	50	43.77
14	30:1	80	50	43.32
15	30:1	80	50	43.14
16	30:1	80	50	42.48
17	30:1	80	50	43.13

**Table 4 foods-12-03438-t004:** ANOVA of RSM for PsP.

Source	Sum of Squares	df	Mean Square	F-Value	*p*-Value	Significance
Model	115.26	9	12.81	52.84	<0.0001	Significant
A-A	36.02	1	36.02	148.64	<0.0001	**
B-B	1.93	1	1.93	7.96	0.0257	
C-C	2.39	1	2.39	9.84	0.0164	
AB	1.38	1	1.38	5.70	0.0483	
AC	1.53	1	1.53	6.33	0.0401	
BC	0.0331	1	0.0331	0.1365	0.7228	
A^2^	26.79	1	26.79	110.53	<0.0001	**
B^2^	30.86	1	30.86	127.35	<0.0001	**
C^2^	7.45	1	7.45	30.75	0.0009	**
Residual	1.70	7	0.2424			
Lack of Fit	0.8371	3	0.2790	1.30	0.3900	Not significant
Pure Error	0.8593	4	0.2148			
Cor Total	119.96	16				

Note: **, *p* < 0.01 represents an extremely significant difference. df: Degree of freedom.

**Table 5 foods-12-03438-t005:** IC50 value for extracts and vitamin C.

Type of Experiment	PsP (mg/mL)	Vitamin C (ug/mL)
Scavenging rate of DPPH radical	20.79	23.3
Scavenging rate of ABTS radical	20.97	9.61
Scavenging rate of hydroxyl radical	118.16	827.22

## Data Availability

The data supporting the reported results can be obtained from the corresponding author.

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
