# Peer review of "A New Method of Extracting Polygonatum sibiricum Polysaccharide with Antioxidant Function: Ultrasound-Assisted Extraction-Deep Eutectic Solvents Method"

_foods, 2023, doi:10.3390/foods12183438_

Round 1
Reviewer 1 Report
Manuscript Title: A new Method of Extracting Polygonatum sibiricum Polysaccharide with antioxidant function: Ultrasound assisted Extraction-Deep eutectic solvents method
The authors reported an efficient method for extraction of polysaccharides from Polygonatum sibiricum, a traditional Chinese food, by using deep eutectic solvent coupled with ultrasound. This efficient extraction can promote its application in the food industry as a natural dietary supplement.
The manuscript is well-written, and the presentation is sound. However, the manuscript demands some minor corrections. I have highlighted a few points below for authors which need to be addressed before acceptance for publication.
1. Line 42-44: Please cite the reference using the Author’s name, not the Journal, and rewrite the sentences.
2. Line 51: “A study of Foods [15] compared the extraction yields of these different extraction methods for PsP” – this study was conducted by Jing et al. (2023). Please give due recognition to the research group.
3. Line 56: “One study [16] once pointed……….” - mention the author
4. While citing a reference use et al. after the first author’s name. Check the manuscript thoroughly.
5. Line 76-77: “….. extraction yield of only ultrasound is slow, it is still possible to use ultrasound……” extraction yield should be low, not slow.
6. Line 95: “…….Polygonatum sibiricum from Beijin……..” is it Beijin or Beijing?
7. Line 100: Complete the sentence
8. The procedure for the preparation of deep eutectic solvent should be clearly mentioned in a separate subheading under the experimental section. The temperature used should be clearly mentioned.
9. Section 2.3: Please provide the detailed method of dialysis procedure including the solvent used.
10. The reference should be cited for the calculation of extraction yield.
11. Section 2.4, Line 140: The appropriate reference should be cited.
12. Line 146-147: Please use space after the comma symbol.
13. Section 2.7: The purchase detail of the “chromatographic column” should be mentioned in the respective section.
14. “Ao: Absorbance of black group”- what is the black group?
15. Line 203: ”….. to prepare ABTS solution”—it should be ABTS free radical.
16. Section 2.9.1-2.9.3: Please mention the units of scavenging activity for clarity.
17. Section 2.9.2. Scavenging rate of ABTS free radical: From where the method has been adopted? Cite a reference.
18. Section 2.9.4. ferric ion reducing ability- A reference for the method needs to be cited. The author may cite this reference- https://pubs.rsc.org/en/content/articlelanding/2021/ra/d0ra09529j
18. Section 2.9.5 and Section 2.9.6 are also missing the reference_ please add appropriate references.
19. The figure resolution should be improved for clear visibility to the readers.
20. Define ‘df’ in the Table 4 footnote.
21. Line 343: “..343 3 that the model P < 0.0001 is extremely significant, indicating that the model is effective ..” – write ‘p’ in lowercase.
22. Correct the peak names in Figure 4.
23. Figure 3 – make necessary corrections.
24. In Figures 6 and 7, write Vitamin C clearly.
25. From the discussion section, Line no 432-438 is not relevant to the current research work and it will be more appropriate in the introduction section. The author may consider moving this part to the introduction section.
26. Please check the bibliography thoroughly and write the scientific names in italics, many times it is missing. For e.g., Ref 9, 16, 27, 28, etc.
27. The Ref 24 could not be understood, if possible, replace it with a different one.
The manuscript is well-written and the presentation is sound.
Author Response
Response to Reviewer 1 Comments
Point 1: Line 42-44: Please cite the reference using the Author’s name, not the Journal, and rewrite the sentences.
Response 1: Thank you very much, and we agree with your suggestion. As your suggestion, we have revised the sentence as follows: “In recent years, the application of Polygonatum sibiricum polysaccharide (PsP) has also received extensive attention, and Wan et al. [8] once reported PsP had been applied to the research of functional food development in 2023, which results showed that PsP had the potential to be a dietary supplement.” The re-edited sentence was added in line 41-45 under the revision mode.
Point 2: Line 51: “A study of Foods [15] compared the extraction yields of these different extraction methods for PsP” – this study was conducted by Jing et al. (2023). Please give due recognition to the research group.
Response 2: Thank you very much, and we agree with your suggestion. As your suggestion, we have revised the sentence as follows: “Jing et al. [15] once compared the extraction yields of these different extraction methods for PsP. This study perfects the basic research on the extraction process of PsP and provides valuable experience for the follow-up research”. The revised part was added in line 52-54 under the revision mode.
Point 3: Line 56: “One study [16] once pointed……….” - mention the author.
Response 3: Thank you very much, and we agree with your suggestion. As your suggestion, we have revised the sentence as follows: ”Kakar et al. [17] once pointed out the main principle of extracting polysaccharide is to enhance the diffusion of polysaccharide from the matrix to the extraction solvents.” Because we have rewrote the Ref.number according to the new sequrence of references, the Ref. number became [17]. The re-edited sentence was added in line 60-62 under the revision mode.
Point 4: While citing a reference use et al. after the first author’s name. Check the manuscript thoroughly.
Response 4: Thank you very much, and we agree with your suggestion. As your suggestion, we have revised this manuscript thoroughly. The re-edited sentence was added in line 43, 52, 60, 82 and 101 under the revision mode.
Point 5: Line 76-77: “….. extraction yield of only ultrasound is slow, it is still possible to use ultrasound……” extraction yield should be low, not slow.
Response 5: Thank you very much, and we agree with your suggestion. As your suggestion, we have revised the sentence as follows: “Although the extraction yield of only ultrasound is low, it is still possible to use ultrasound as an assistance condition to improve the extraction yield of other methods, because the material matrix is more easily broken, enabling efficient and short-term release of polysaccharide under the conditions of ultrasound assistance [29].” The revised part was added in line 102-106 under the revision mode.
Point 6: Line 95: “…….Polygonatum sibiricum from Beijin……..” is it Beijin or Beijing?
Response 6: Thank you very much, and we agree with your suggestion. As your suggestion, we have revised the sentence as follows: “As the research for PsP in North China was not enough, we specifically selected P.s from Beijing as the source material in our research.” The revised part was added in line 120-121 under the revision mode.
Point 7: Line 100: Complete the sentence
Response 7: Thank you very much, and we agree with your suggestion. As your suggestion, we have completed the sentence as follows: “The roots of P.s were selected from Shunyi District (Beijing, China). All other chemicals were of analytical grade. All the reagents of this research can be searched from Table 1.” The completely sentence was added in line 129-131 under the revision mode.
Point 8: The procedure for the preparation of deep eutectic solvent should be clearly mentioned in a separate subheading under the experimental section. The temperature used should be clearly mentioned.
Response 8: Thank you very much, and we agree with your suggestion. As your suggestion, we have added a separate subheading including the introductions of procedure and clearly temperature as follows: ”Choline chloride and 1,4-butanediol were mixed in a molar ratio of 1:4, then a certain amount of water was added to make the water content reach 33%. The water bath was stirred until completely dissolved at 60℃ until a clear and transparent deep eutectic solvent was formed. ” The procedure for the preparation of DESs was added in Section 2.4 (In line 165-169) under the revision mode.
Point 9: Section 2.3: Please provide the detailed method of dialysis procedure including the solvent used.
Response 9: Thank you very much, and we agree with your suggestion. As your suggestion, we have revised the sentence as follows: “The dialysis bag with 1000Da specification was selected for dialysis. Put PsP in a dialysis bag and put it on a magnetic agitator and rotate in distilled water for dialysis. Changed the distilled water every 24 hours for a total of 2 days. The impurities and small molecules were further removed by centrifugation after dialysis.” The detailed method of dialysis procedure was added in line 185-189 under the revision mode.
Point 10: The reference should be cited for the calculation of extraction yield.
Response 10: Thank you very much, and we agree with your suggestion. As your suggestion, we have added reference [32] about calculation of extraction yield. The added reference was in line 191 and “References section” (In line 843-844 ) under revision mode.
Point 11: Section 2.4, Line 140: The appropriate reference should be cited.
Response 11: Thank you very much, and we agree with your suggestion. As your suggestion, we have replaced an appropriate reference in this section. The added reference [33] was in line 201 and “References section” (In line 845-847) under revision mode.
Point 12: Line 146-147: Please use space after the comma symbol.
Response 12: Thank you very much, and we agree with your suggestion. As your suggestion, we have used space after the comma symbol: “The effects of liquid-solid ratio [10:1, 20:1, 30:1, 40:1, 50:1 (mL: g)], ultrasonic power (40, 60, 80, 100, 120W), ultrasonic time (10, 20, 30, 40, 50, 70min) and extraction temperature (40, 50, 60, 70, 80℃) on the extraction yield of PsP and the scavenging rate of DPPH radicals were investigated, respectively.” The added space was in line 203-206 under revision mode.
Point 13: Section 2.7: The purchase detail of the “chromatographic column” should be mentioned in the respective section.
Response 13: Thank you very much, and we agree with your suggestion. As your suggestion, we have added the purchase detail of column information in Section 2.2 (In line 148-152 of Table 2).
Point 14: “Ao: Absorbance of black group”- what is the black group?
Response 14: Thank you very much for your professional question. I’m very sorry for this spelling mistake. We want to express “blank group” in this sentence, but not black group. We have revised the sentence as follows: Ao: Absorbance of blank group, revised sentence was in line 286 under revision mode.
Point 15: Line 203: ”….. to prepare ABTS solution”—it should be ABTS free radical.
Response 15: Thank you very much, and we agree with your suggestion. As your suggestion, we have revised the sentence as follows: 7 mmol/L ABTS methanol solution and 2.45 mmol/L potassium persulfate solution were mixed at 1:1 and reacted at 4 ℃ for 24 h to prepare ABTS free radical. The re-edited part was in line 288-290 under revision mode.
Point 16: Section 2.9.1-2.9.3: Please mention the units of scavenging activity for clarity.
Response 16: Thank you very much, and we agree with your suggestion. As your suggestion, we have mentioned the units of scavenging activity. Because we have rewrote the section number according to the new sequrence of section, the section numbers of “2.9.1-2.9.3” became “2.11.1-2.11.3”. The added units of scavenging activity were in section 2.11.1-2.11.3 (line 283, 296 and 307) under revision mode.
Point 17: Section 2.9.2. Scavenging rate of ABTS free radical: From where the method has been adopted? Cite a reference.
Response 17: Thank you very much, and we agree with your suggestion. As your suggestion, we have added reference [37] of scavenging rate of ABTS free radical. The added reference was in line 294 and “References section” (In line 858-859) under revision mode.
Point 18: Section 2.9.4. ferric ion reducing ability- A reference for the method needs to be cited. The author may cite this reference- https://pubs.rsc.org/en/content/articlelanding/2021/ra/d0ra09529j
Response 18: Thank you very much, and we agree with your suggestion. As your suggestion about https://pubs.rsc.org/en/content/articlelanding/2021/ra/d0ra09529j, we have added reference [38] of ferric ion reducing ability for that. The added reference [38] was in line 324 and “References section” (In line 860-861) under revision mode.
Point 19: Section 2.9.5 and Section 2.9.6 are also missing the reference_ please add appropriate references.
Response 19: Thank you very much, and we agree with your suggestion. As your suggestion, we have added references [39] and [40] for them. The added references were in line 343, 353 and “References section” (In line 862-863, 864-865) under revision mode.
Point 20: The figure resolution should be improved for clear visibility to the readers.
Response 20: Thank you very much, and we agree with your suggestion. As your suggestion, we have improved the clarity of each figure in the whole manuscript. The revised figures were in line 402, 472-474, 491-492, 500-501, 521-522, 538-541, 555-557 and 566-567 under revision mode.
Point 21: Define ‘df’ in the Table 4 footnote.
Response 21: Thank you very much, and we agree with your suggestion. The df means “Degree of freedom”. As your suggestion, we have added the definition of ‘df’ in the Table 5 footnote (In line 450) under the revision mode.
Point 22: Line 343: “..343 3 that the model P < 0.0001 is extremely significant, indicating that the model is effective ..” – write ‘p’ in lowercase.
Response 22: Thank you very much, and we agree with your suggestion. As your suggestion, we have replaced 'P' to 'p’ in lowercase. The re-edited part was in line 452 under the revision mode.
Point 23: Correct the peak names in Figure 4.
Response 23: Thank you very much, and we agree with your suggestion. As your suggestion, we have corrected the peak names in Figure 4. The revised part was in line 500-501 under revision mode.
Point 24: Figure 3 – make necessary corrections.
Response 24: Thank you very much, and we agree with your suggestion. As your suggestion, we have revised the Figure 3 to make it more clear for readers to understand. The revised figure was in line 491-492 under revision mode.
Point 25: In Figures 6 and 7, write Vitamin C clearly.
Response 25: Thank you very much, and we agree with your suggestion. As your suggestion, we have re-edited the Vitamin C in the Figure 6 and 7. The revised figures were in line 538-541 and 555-557 under revision mode.
Point 26: From the discussion section, Line no 432-438 is not relevant to the current research work and it will be more appropriate in the introduction section. The author may consider moving this part to the introduction section.
Response 26: Thank you very much, and we agree with your suggestion. As your suggestion, we have moved this part from discussion section to the introduction section. The revised part was in line 71-75 under revision mode.
Point 27: Please check the bibliography thoroughly and write the scientific names in italics, many times it is missing. For e.g., Ref 9, 16, 27, 28, etc.
Response 27: Thank you very much, and we agree with your suggestion. As your suggestion, we have revised the format as scientific names in italics in the bibliography thoroughly. Because we have rewrote the Ref.number according to the new sequrence of references, the revised references were in line 760 (Ref.1), 762 (Ref.2), 764 (Ref.3), 768 (Ref.4), 774 (Ref.7), 778 (Ref.8), 780 (Ref.9), 784 (Ref.10), 789 (Ref.12), 795-796 (Ref.15), 798 (Ref.116), 801 (Ref.17), 807 (Ref.19), 833 (Ref.28), 836 (Ref.29), 854 (Ref.35), 866 (Ref.41), 869 (Ref.42) and 873 (Ref.44) under the revision mode.
Point 28: The Ref 24 could not be understood, if possible, replace it with a different one.
Response 28: Thank you very much for your patience. We arer very sorry about Ref 24, because it is our mistake in expression. We have revised the information of the Ref 24 and rewrote the Ref.number according to the new sequrence of references. The rewrote Ref number [25] and the correct expression are in line 82, 167, 597, 600, 631,823-826 under the revision mode.
Above information are our response for your reviews. Thank you very much, and your suggestions have absolutely improved the quality of our article, and we hope the revised contents can be accepted as your requirement. We hope we can obtain your support for our research, and we can conduct more cooperation with you in polysaccharide research in future. Thank you very much again.
Yours Sincerely
The whole members of PsP research team

Reviewer 2 Report
The research article “A new Method of Extracting Polygonatum sibiricum Polysaccharide with antioxidant function: Ultrasound assisted Extraction-Deep eutectic solvents method” focuses on obtaining polysaccharide fraction with good antioxidant activity by ultrasound-assisted extraction-deep eutectic solvents (UAE-DESs) method. This paper may be suitable for publication following major revision. The authors should answer and resolve the following queries.
1. When the plant is mentioned for the first time, the full Latin name be written and should be written in italics, but after that, there is no need to write the full name through the whole manuscript.
2. “In vitro” should be written in italics. Please, correct in whole paper.
3. Why is a mix of Choline chloride and 1,4 butanediol chosen as a deep eutectic solvent?
4 . In line 35 space should be added before reference “applied[2]”. The space is missing after interpunction signs throughout the whole manuscript. Please, correct this.
5. In line 39 is mentioned “hypoglycemic activity”. Please, check and add a reference for this if this reference is mentioned at the end.
6. Please check Reference 24
7. Please add an abbreviation list and write and full name and abbreviation when something is mentioned for the first time in the paper (UP, TFA)
8. Only assays based on in vitro antioxidant capacity measurements were used for testing antioxidant activity. This type of assay has many drawbacks. In this paper, the results are not clearly presented or explained.
9. Why is the antioxidant assay used formula
DPPH radical scavenging rate =(1 - (Aj-Ai)/Ao)×100 (2)
Aj: Absorbance of PsP,
Ai: Absorbance of control group,
Ao: Absorbance of black group.
Why is not inhibition calculated? Which concentration of PsP was used?
Which concentration of vitamin C is used as the positive control?
10. FRAP – comparison with positive control?
11. Catalase activity determination in vitro –method should be better explained
12. Resolutions of the graphics in results are very bad.
13. The results of the antioxidant assay are not clearly presented.
It would be better that in one table presented the IC50 value for extracts and vitamin C, as positive control.
Why are the results of the FRAP assay expressed on the graphic as FRAP value?
14. In lines 468-471 is written “Under above conditions, the extraction yield of PsP could reach 43.61%, which is obviously higher than 34.26% of only using deep eutectic solvents and 5.83% of only using ultrasound-assisted extraction in extracting PsP [8,24] .”
Please, check the reference. A yield of 5.83% is mentioned in reference 15.
What were other parameters used in the assay with which is compared (solid: liquid ratio, ultrasonic power and so on)?
15. How is confirmed which monosaccharides are in PsP?
Moderate editing of English language is required.
Author Response
Response to Reviewer 2 Comments
Point 1: When the plant is mentioned for the first time, the full Latin name be written and should be written in italics, but after that, there is no need to write the full name through the whole manuscript.
Response 1: Thank you very much, and we agree with your suggestion. As your suggestion, we have added the Latin name (Polygonati Rhizoma) of Polygonatum sibiricum which was mentioned for the first time (In Line 32 under the revision mode), and we used P.s as abbreviation of Polygonatum sibiricum in line 13, 32, 33, 121, 126, 129, 171, 172, 174, 183, 196, 670 and 733 through the whole manuscript under the revision mode.
Point 2: “In vitro” should be written in italics. Please, correct in whole paper.
Response 2: Thank you very much, and we agree with your suggestion. As your suggestion, we have revised the format of “In vitro” in whole paper. The revised “In vitro” were in line 19, 275, 346, 559, 564, 568, 658, 660, 664, 680, 692, 693 and 696 under the revision mode.
Point 3: Why is a mix of Choline chloride and 1,4 butanediol chosen as a deep eutectic solvent?
Response 3: Thank you very much for your professional question. The reason that we chosed the mixture of choline chloride and 1,4 butanediol as deep eutectic solvents is:
“Deep eutectic solvents comprise different hydrogen-bonded donors (1,4 butanediol, urea, glycerol, citric acid, etc.) and receptors (choline chloride, betaine, etc.), resulting in distinct characteristics among various deep eutectic solvents. In this research, we selected the deep eutectic solvent composed of choline chloride and 1,4 butanediol, which based on the results of numerous comparative studies on the extraction yield of polysaccharides among different types of deep eutectic solvents [25]. The comparative results showed the deep eutectic solvents made from the mixture of choline chloride and 1,4 butanediol demonstrated a significantly higher extraction yield of polysaccharides than other deep eutectic solvents [25]. The main reason is deep eutectic solvent composed of choline chloride and 1,4 butanediol may be more suitable for the extraction of PsP according to the principle of formulation design of deep eutectic solvents [18, 21]. Given this, we selected the mixture of choline chloride and 1,4 butanediol as deep eutectic solvents to combine with ultrasound in the subsequently research. The result showed that the combination of ultrasound and deep eutectic solvents had actually achieved the improvement of the polysaccharide extraction efficiency.”
The above introduction have been added in lin 592-603 of the revised manuscript under the revision mode.
Point 4: In line 35 space should be added before reference “applied[2]”. The space is missing after interpunction signs throughout the whole manuscript. Please, correct this.
Response 4: Thank you very much, and we agree with your suggestion. As your suggestion, we have revised this missing in line 36 and 622 under the revision mode.
Point 5: In line 39 is mentioned “hypoglycemic activity”. Please, check and add a reference for this if this reference is mentioned at the end.
Response 5: Thank you very much, and we agree with your suggestion. As your suggestion, we have checked and added reference [5] for “hypoglycemic activity” in line 40, 766-768 (References section) under the revision mode.
Point 6: Please check Reference 24
Response 6: Thank you very much for your patience. We are very sorry about Ref 24, because it is our mistake in expression. We have revised the information of the Ref 24 and rewrote the Ref.number according to the new sequrence of references. The rewrote Ref number [25] and the correct expression are in line 82, 167, 597, 600, 662, 631,823-826 (References section) under the revision mode.
Point 7: Please add an abbreviation list and write and full name and abbreviation when something is mentioned for the first time in the paper (UP, TFA)
Response 7: Thank you very much, and we agree with your suggestion. As your suggestion, we have added an abbreviation list (Table 1) and wrote and full name and abbreviation when something is mentioned for the first time in the manuscript (In line 126-127) under the revision mode..
Point 8: Only assays based on in vitro antioxidant capacity measurements were used for testing antioxidant activity. This type of assay has many drawbacks. In this paper, the results are not clearly presented or explained.
Response 8: Thank you very much, and we agree with your opinions to more clearly express the in vitro antioxidant experiments, and to more clearly present the results. In the revised manuscript, we added below introductions.
“Although in vitro antioxidant experiments have limitations, they can still be a necessary, important and indispensable section in the whole process of the antioxidant evaluation. Therefore, there were still many scholars adopted in vitro antioxidant experiments as a necessary part of their researches, which can provide a necessary exploration for the follow-up in vivo antioxidant experiments [25, 36, 38]. In our study, we utilized six different indexes to analyze and compare the antioxidant activity of PsP, so as to more comprehensively reduce the system error caused by the limitations of in vitro antioxidant experiments as possible, and extremely prove the antioxidant activity of PsP in this section. This study did not attempt to use in vitro antioxidant experiments to replace in vivo antioxidant experiments, and the in vitro antioxidant experiment is only the starting point of our team in proving the antioxidant activity of PsP, but not the end point of that. In the follow-up studies, we will successively carry out the in vivo antioxidation experiment of PsP based on the above results of in vitro antioxidant experiments, so as to more comprehensively explore the relevant mechanism and pathway of PsP antioxidation, and provide solid foundation for PsP as a dietary supplement in the food industry.
The results of the DPPH, ABTS and hydroxyl scavenging experiments in this study showed that the scavenging effect of PsP on radicals increased gradually with concentrations ranging from 1-16mg/mL, with a more pronounced effect observed at higher concentrations. The reasons might be the polysaccharide molecules of P.s have strong hydrophilicity and can form a large number of molecular chains in water [45], and these molecular chains can provide electrons to the radicals, so as to complete the unstable electrons in the radicals and make them stable, thus achieving the effect of scavenging radicals. Similarly, the ferric ion reducing ability and total antioxidant capacity experiments demonstrated a gradual enhancement of antioxidant activity with an increase of PsP concentration (1-16mg/mL). Moreover, our experiment about the effect of PsP on in vitro catalase activity indicated that the activity of catalase could be promoted by PsP at concentration range of 1-4mg/mL. However, with the continuous increase of the concentration of PsP, the promoting effect of PsP on catalase activity gradually weakened after 4 mg/mL, and even showed inhibitory effect. This phenomenon may be attributed to the low concentration of PsP may provide an enviornment which can improve the activity of catalase and make the catalase more stable and active. However, the over high concentration of PsP would alter the pH environment of catalase, and then, induce structural changes of that, which would lead to the decrease of catalase activity. This finding suggested that PsP may enhance the catalase activity within a specific concentration range, and provided a novel pathway for future research on the effects of PsP on catalase activity. The above results of antioxidant activity experiments were consistent with other similar research [16,18].”
The relevant results of follow-up research about in vivo antioxidant experiments will also continuously be submitted to Foods journal. The revised part was adde in line 658-698 under the revision mode. The clear results were presented or explained in line 528-531, 547-550 and 560-563 of “Results section” under the revision mode.
Point 9: Why is the antioxidant assay used formula
DPPH radical scavenging rate =(1 - (Aj-Ai)/Ao)×100 (2)
Aj: Absorbance of PsP,
Ai: Absorbance of control group,
Ao: Absorbance of black group.
Why is not inhibition calculated? Which concentration of PsP was used?
Which concentration of vitamin C is used as the positive control?
Response 9: Thank you very much for your professional questions and suggestions. I’m very sorry for our mistake about “Ao” in this formula. We want to express “blank group” in this sentence, but not black group. We have revised the sentence as follows: Ao: Absorbance of blank group, and the revised sentence was in line 286 under revision mode.
The reason why the scavenging rate of DPPH radical is used as the calculation formula is that the scavenging rate can directly reflect the scavenging effect of the substance in scavenging DPPH radical, which is a simple and effective evaluation method [36]. DPPH is a dark purple stable radical with strong absorption properties, and its absorbance is proportional to its concentration. When a substance has antioxidant capacity, it will donate a hydrogen atom or electron to the DPPH radical, neutralizing its instability and turning it into a yellowish color. Therefore, the scavenging rate of the sample to DPPH radical can be calculated by measuring the absorbance of the sample after reacting with DPPH radical. The formula used in this research is also referred to Ref. 36 (In line 279, 856-857), and combined with our own experimental needs, so the formula is used in our research.
For the second question, the reason is the calculations of the scavenging rate and inhibition rate of DPPH radicals are same, because the principles both of the two calculations are that antioxidants could provide hydrogen atoms or electrons to DPPH radicals, and then transform DPPH radicals into stable molecules. Both the inhibition rate and radical scavenging rate expressed the same result through different angles. In this experiment, both the inhibition rate and radical scavenging rate can be used to express the antioxidant activity of PsP. Therefore, the scavenging rate and inhibition rate of DPPH radicals are calculated in the same way, that is, both the scavenging and inhibitory ability of antioxidants to DPPH radicals are evaluated by comparing the difference of light absorption of DPPH radicals between the experimental group and the control group. In order to express the calculated results of inhibition rate as your suggestion, we increase the IC50 value of PsP on DPPH radical in Table 6 (line in 543-545) under the revision mode.
For the third question, we agree with you to specify the concentrations of PsP and vitamin C in the introduction of the methods in the experiment of DPPH radical scavenging rate and other experiment. In the experiment of DPPH radical scavenging rate, the concentrations of PsP included 1, 2, 4, 8, 16 mg/mL, and the concentrations of vitamin C included 10, 20, 40, 60, 80 μg/mL. As your suggestions, the concentrations of PsP and vitamin C were added in the line 278, 280, 291, 294, 302, 304, 317, 321, 336, 342 and 348-349 under the revision mode.
For the fouth question, the concentrations of vitamin C of positive control included 10, 20, 40, 60, 80 μg/mL. The above concentrations of vitamin C were added in the in line 280 under the revision mode. The reason why we chose multiple concentrations of vitamin C as the positive control was we wanted to prove the reliability of the scavenging test of DPPH radical through the dose-response relationship between vitamin C and the scavenging rate of DPPH radicals, so as to demostrate the antioxidant activity of PsP through observed whether PsP could appeared a similar dose-response relationship to the scavenging rate of DPPH radicals.
Point 10: FRAP – comparison with positive control?
Response 10: Thank you very much for your comments. We agree with your suggestion to more clearly introduce the process of the method about positive control and the display of the results.
In this experiment, we used the FRAP value to reflect the antioxidant activity of the tested samples. At the same time, we compared the FRAP results between PsP and positive control (vitamin C ) in different concentrations and showed them by drawing point plot. In order to more clearly introduce the process of the method about positive control and the display of the results. We have re-edited the method as follows: “Tripyridyltriazine solution was prepared with dilute hydrochloric acid of 40 mmol/L concentration and its terminal concentration was 10 mmol/L. Sodium acetate solution of 300 mmol/L, FeCl3 solution of 20 mmol/L and tripyridyltriazine solution were mixed according to volume ratio of 10:1:1 to prepare tripyridyltriazine working solution (now used). PsP solution and ferrous sulfate standard solution of 0.1 - 0.6 mmol/L were prepared. 0.3 mL PsP solutions (1, 2, 4, 8, 16 mg/mL) and different concentrations of standard solution were mixed with 2.7 mL tripyridyltriazine working solution, shaken well, water bathed at 37 ℃ for 10 min and the absorbance was measured at 593 nm by spectrophotometer. At the same time, different concentrations of vitamin C solution (0.625, 1.25, 2.5, 5, 10 μg/mL) were used as positive control of PsP to conduct above experiments. Taking the concentration of ferrous sulfate standard solution as Ordinate and absorbance value as Abscissa, and then the standard curve of ferrous sulfate was drawn [38]. Through measured the average absorbance, the corresponding Fe2SO4 concentration was obtained on the standard curve, and the results express by ferrous sulfate equivalent, namely FRAP value (μmol/g) of the sample.” The revised part was in Methods 2.11.4 (Line in 312-326) under the revision mode. At the same time, we have re-edited the display of result in line 547-550, and the revised point plot are shown in Figure 7. (In line 555-556) under the revision mode.
Point 11: Resolutions of the graphics in results are very bad.
Response 11: Thank you very much, and we agree with your suggestion. As your suggestion, we have improved the resolution of the graphics (Figre 3, 4, 5, 7 ,8) in the results under the revision mode(In line 491-492, 500-501, 521-522, 555-557 and 566-567).
Point 12: The results of the antioxidant assay are not clearly presented.
It would be better that in one table presented the IC50 value for extracts and vitamin C, as positive control.
Why are the results of the FRAP assay expressed on the graphic as FRAP value?
Response 12: Thank you very much, and we agree with your suggestion. As your suggestion, we have used the IC50 value to present the results of DPPH, ABTS and hydroxyl radical scavenging rates of PsP and vitamin C to conduct the antioxidant assay more clearly in Table 6 (line in 543) under the revision mode.
“In the experiment of ferric ion reducing ability, we took the concentration of ferrous sulfate standard solution as Ordinate and absorbance value as Abscissa, and the standard curve of ferrous sulfate was drawn”. After the absorbance of the PsP and vitamin C were detected, the concentration of ferrous sulfate is calculated by the standard curve. The antioxidant capacies of the PsP and vitamin C are expressed by ferrous sulfate equivalent, namely FRAP value (μmol/g). “The FRAP value can better show the results of antioxidant activity, and many researchers like to use this index to present the antioxidant activity [38]”. Above the informations were added in line 322-326, 676-679 under the revision mode.
In order to enable readers can understand the research content more clearly, we have added a reference [38] in line 324, 860-861 under the revision mode.
Point 13: In lines 468-471 is written “Under above conditions, the extraction yield of PsP could reach 43.61%, which is obviously higher than 34.26% of only using deep eutectic solvents and 5.83% of only using ultrasound-assisted extraction in extracting PsP [8,24] .”
Please, check the reference. A yield of 5.83% is mentioned in reference 15.
What were other parameters used in the assay with which is compared (solid: liquid ratio, ultrasonic power and so on)?
Response 13: Thank you very much for your patiences, and we agree with your suggestion. This is our mistake when we quote the literature. As your suggestion, we have revised the number references of Ref.8 in the old manuscript. After we revised the sequrence of references, the new number about the yield of 5.83% is mentioned in reference [16], which is in line 52, 101, 631, 691 and 796-798 under the revision mode.
For introducing other parameters used in the assay with which is compared, we have modified the sentence to add the experimental parameters of the other two experiments as follow:”Under above conditions, the extraction yield of PsP could reach 43.61%, which is obviously higher than 33.81% of only using deep eutectic solvents (solid:liquid ratio is 1:40, extraction time is 48 min, extraction temperture 89℃) and 5.83% of only using ultrasound-assisted extraction (solid:liquid ratio is 1:30, extraction time is 40 min, extraction temperture 60℃, ultrasonic power is 100W) in extracting PsP.”
We have added the above parameters in line 626-631 under the revision mode.
Point 14: How is confirmed which monosaccharides are in PsP?
Response 14: Thank you very much for your professional question. I’m sorry about our unclear introduction for this method in the article, and we have re-edited this section according to your question, as follows: “Monosaccharide mixed reference solution included: D-mannose, D-ribose, L-rhamnose, D-glucuronic acid, D-galacturonic acid, D-glucose, D-galactose, D-xylose, D-arabinose and L-fucose, and weighed each one of them precisely, respectively, and added water to prepare the monosaccharide mixed reference solution (the terminal concentration of each monosaccharide is 0.1 g / mL). Weighed a proper amount of the PsP in the ampoule bottle, added 1 mL UP water to fully dissolve it, then added 1 mL 4 M TFA. The mixed solution was hydrolyzed with trifluoroacetic acid (2 M) at 110 ℃ for 4 h in the ampoule bottle. Opened the ampoule bottle and sucked out the sample solution, neutralized it with NaOH and volumed to 10 mL with distilled water. Took 200 μL constant volume solution and added 200 μL internal standard (2-deoxyribose), and the mixture was shaken and mixed well. 100 μL of the mixture was absorbed into a 1.5 mL centrifuge tube, then added 120 μL of 0.3 M NaOH and 0.5 M PMP, respectively. Mixed the mixture thoroughly and took a water bath at 70 ℃ for 1 hour, avoided light. After water bath, the centrifuge tube was cooled down to room temperature, then added 100 μL 0.3 M HCI to neutralize the mixed solution to a pH of 7. Added 500 μL chloroform to the centrifuge tube for extraction, mixed thoroughly and centrifuged (7000 rpm × 5 min). The upper solution (water layer) was carefully sucked out, and transferred it into a new centrifuge tube. This operation was repeated for 4 times. The supernatant extracted from the last extraction was filtered into a liquid phase bottle by using a 0.22 um filter membrane, and conducted HPLC analysis on the liquid in the liquid phase bottle. The detection conditions were as follows: the chromatographic column was Agilent Eclipse XDB-C18 (5 μm, 4.6 × 250 mm), the flow rate was 1 mL/min, the column temperature was 30 ℃, the detection wavelength was 254 nm, the mobile phase was 18% acetonitrile triethylamine solution and the mobile phase was 60% acetonitrile triethylamine solution. Made a comparison of the HPLC maps between PsP and mixed monosaccharides reference solution, and used 2-deoxyribose as the internal standard. Finally, the molar ratio of monosaccharide composition was calculated by internal standard method [35]. “The revised part was added in Materials and Methods 2.10 (In line 230-257) under the revision mode, and we have changed the subtitle to the monosaccharide component detection in line 230 under the revision mode.
At the same time, we have also re-edited the result display of this part, as follows: “Comparing the HPLC maps between PsP and mixed monosaccharides reference solution, it was found that five peaks in PsP were consistent with those of the mixed monosaccharides reference solution, indicating that PsP was composed of five monosaccharides. Depended on the peak position of the mixed monosaccharides reference solution, it was known that the five monosaccharides of PsP were respectively mannose, glucuronic acid, glucose, galactose and xylose. By using 2-deoxyribose as the internal standard as internal standard method, the molar ratios of monosaccharide composition of them were 0.86, 0.30, 1.00, 0.87, 0.05, respectively.” The revised part was added in Result 3.4 (In line 508-516) under the revision mode.
Above information are our response for your reviews. Thank you very much, and your suggestions have absolutely improved the quality of our article, and we hope the revised contents can be accepted as your requirement. We hope we can obtain your support for our research, and we can conduct more cooperation with you in polysaccharide research in future. Thank you very much again.
Yours Sincerely
The whole members of PsP research team

Reviewer 3 Report
This is an interesting subject, and the manuscript is well-prepared. However, I doubt that deep eutectic solvents could be successfully applied in the food industry. They could not be easily removed. There are safety issues as well as the complexity of manipulation.
Author Response
Response to Reviewer 3 Comments
Point 1: However, I doubt that deep eutectic solvents could be successfully applied in the food industry. They could not be easily removed. There are safety issues as well as the complexity of manipulation.
Response 1: Thank you very much for your suggestions. I’m sorry about our unclear introduction for the safety issues, the complexity of manipulation, and the successfully application of DESs.
We have re-edited this section according to your question, as follows: “Deep eutectic solvents (DESs), which could enhance the diffusion of polysaccharide, are a novel type of extraction solvents that offer several advantages over conventional solvents and ionic liquids, including low price, easy to obtain, good biodegradability, stability, low toxicity or non-toxic [18-19]. Moreover, the internal structure of DESs can form hydrogen bonds with target polysaccharide and create electrostatic interactions, significantly improving the extraction efficiency of polysaccharide [20-22]. More and more studies have shown that the extraction with DESs of plant active components can not only significantly improve the extraction yield, but also retain the functional activity of the active substances [21]. DESs are characterized by their low surface tension and viscosity, which enable they can penetrate into polysaccharide more effectively and enhance the extraction efficiency. Due to the differences of the high polarity and hydrophobicity of different DESs, they can extract various types of polysaccharides [18]. In the polysaccharide extraction process, ethanol precipitation method was used to obtain polysaccharides. During the ethanol precipitation process, the DESs was dissolved in ethanol, then removed with ethanol by cetrifuge. In the extraction process of polysaccharides, DESs only played the role of extraction solvents without any complex operation. Thus, DESs present a unique advantage, such as safety, easy manipulation and removement, in the extraction and separation of natural plants [23]. As an economical and innovative green solvent, they have been widely applied in the separation, extraction and synthesis of food industries, based on above characteristics [19, 24]. Tang [25] once adopted DESs to extract PsP, which extraction yield could reach to 33.81%.” Above the information were added in the line 63-83 under the revision mode.
At the same time, we have re-edited the introduction about the removement effect of DESs in dicussion, so as to response your question, as follow: “The results of our research showed that UAE-DESs method had improved the PsP extraction yield to 43.61% and preserved its antioxidant function to the greatest extent, the purity of PsP reached 98.76%, and the contents of nucleic acid and protein only were 0.78 ± 0.04% and 1.11 ± 0.32%, respectively, so it can be considered that the DESs had been removed. Above results showed that UAE-DESs method could meet the requirements of industrial production of PsP, and the PsP that was extracted from UAE-DESs method could be developed and applied as dietary supplement in the food industry.” The above contents were added in line 571-578 under the revision mode.
Above information are our response for your reviews. Thank you very much, and your suggestions have absolutely improved the quality of our article, and we hope the revised contents can be accepted as your requirement. We hope we can obtain your support for our research, and we can conduct more cooperation with you in polysaccharide research in future. Thank you very much again.
Yours Sincerely
The whole members of PsP research team

Round 2
Reviewer 3 Report
Thank you for your kind reply to my concerns. After revision of the manuscript, I agree to acceptance of the manuscript in its current form.